# Colliding heavy nuclei take multiple identities on the path to fusion

Kaitlin J. Cook [1,2] ✉, Dominic C. Rafferty[1], David J. Hinde[1], Edward C. Simpson[1], Mahananda Dasgupta[1], Lorenzo Corradi[3], Maurits Evers[1], Enrico Fioretto[3], Dongyun Jeung [1], Nikolai Lobanov [1], Duc Huy Luong[1], Tea Mijatović [4], Giovanna Montagnoli[5], Alberto M. Stefanini[3] & Suzana Szilner[4]

The properties of superheavy elements probe extremes of physics and chemistry. They are synthesised at accelerator laboratories using nuclear fusion, where two atomic nuclei collide, stick together (capture), then with low probability evolve to a compact superheavy nucleus. The fundamental microscopic mechanisms controlling fusion are not fully understood, limiting predictive capability. Even capture, considered to be the simplest stage of fusion, is not matched by models. Here we show that collisions of $^{40}$Ca with $^{208}$Pb, experience an 'explosion' of mass and charge transfers between the nuclei before capture, with unexpectedly high probability and complexity. Ninety different partitions of the protons and neutrons between the projectile-like and target-like nuclei are observed. Since each is expected to have a different probability of fusion, the early stages of collisions may be crucial in superheavy element synthesis. Our interpretation challenges the current view of fusion, explains both the successes and failures of current capture models, and provides a framework for improved models.

The synthesis of new elements is achieved in nuclear collisions, in which two nuclei come close to each other and merge to form a new compound nucleus. The two nuclei must come close enough for their matter distributions to overlap, allowing the attractive nuclear force to act. Then their kinetic energy is rapidly dissipated, and the can system transition to a single, compact, excited nucleus. Preventing their merger (fusion) is a potential barrier, created by the sum of the long-range repulsive Coulomb potential and short-range attractive nuclear potential (Fig. 1a), having a peak at radial separation $R_B$. For fusion to occur, this barrier must be overcome, either by having sufficient energy to pass over it, or through it or via quantum tunnelling. Barrier-passing models of fusion construct this barrier and apply boundary conditions inside $R_B$ to simulate fusion, assuming that the identities (i.e. their proton and neutron numbers) of the nuclei are essentially unchanged prior to this point.

A wealth of experimental results suggests that this picture is too simple. Fusion cross-sections measured very far below the barrier energy are smaller than model[1,2] predictions[3–6]. Suggested explanations are nuclear incompressibility through Pauli repulsion[7,8] or neck formation[9] acting to widen the barrier. Measured above-barrier cross-sections are also systematically smaller than model calculations[6,10]. Experiments suggest that there may be a dynamical origin linking both energy domains—arising from the gradual loss of kinetic energy (energy dissipation) already outside the barrier, before the nuclei touch[6]. Significant kinetic energy loss will lead to substantially reduced cross-sections[6,10,11]. Understanding the early stages of the collision, and the state of the system at the point of fusion, is therefore key.

We cannot directly probe the system as it evolves towards fusion: the transition from isolated nuclei to a compound system is too fast, occurring on a $10^{-21}$ s timescale. However, we can probe the system by

[1]Department of Nuclear Physics and Accelerator Applications, Research School of Physics, The Australian National University, Canberra 2601 ACT, Australia. [2]Facility for Rare Isotope Beams, Michigan State University, East Lansing 48824 MI, USA. [3]Istituto Nazionale di Fisica Nucleare, Laboratori Nazionali di Legnaro, Legnaro I-35020, Italy. [4]Ruđer Bošković Institute, Zagreb HR-10001, Croatia. [5]Dipartimento di Fisica e Astronomia, Universita di Padova, via Marzolo 8, Padova I-35131, Italy. ✉e-mail: kaitlin.cook@anu.edu.au

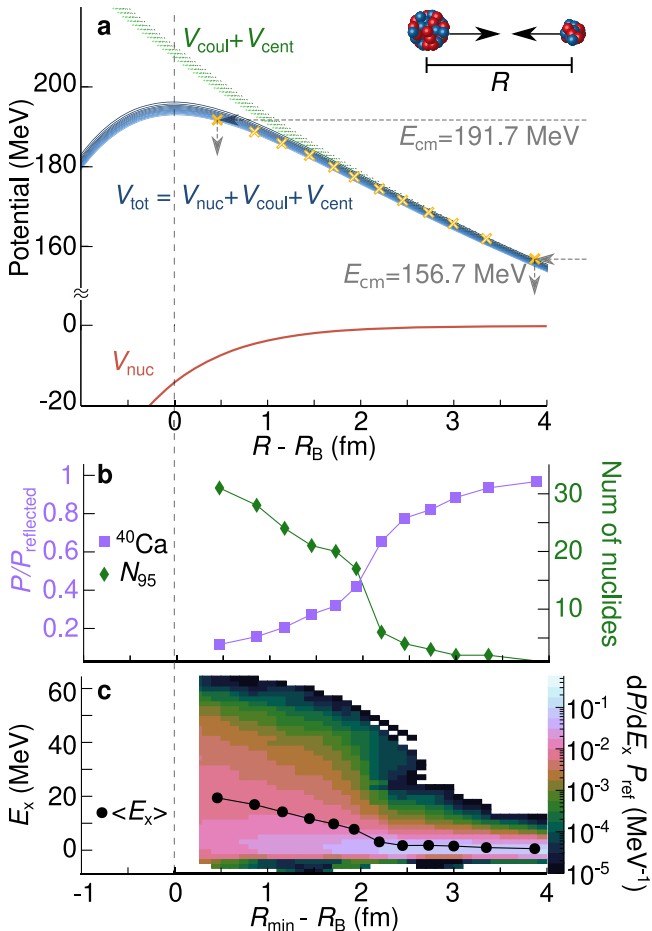

**Fig. 1 | Experimental principle and onset of complexity in reaction outcomes.** **a** Depicted is the internuclear potential (blue) of $^{40}$Ca+$^{208}$Pb nuclei (with centre-to-centre separation $R$), the nuclear potential $V_{\text{nuc}}$ (red)[52], and the sum of the Coulomb and centrifugal potentials $V_{\text{coul}} + V_{\text{cent}}$ (green). The yellow crosses show the deduced distance between the barrier $R_B$ and the distance of closest approach $R_{\text{min}}$, $R_{\text{min}} - R_B$, at each measured energy $E_{\text{cm}}$, taking into account the change in angular momentum $l$ at the measurement angle. **b** Left axis: proportion of the reflected flux $P/P_{\text{reflected}}$ that is made up of $^{40}$Ca (lilac squares), statistical errors are smaller than the points. Right axis: The smallest number of nuclide pairs required to make up 95% of the reflected flux ($N_{95}$, green diamonds). **c** The deduced excitation energy distribution $E_x$. To show the evolution of the excitation energy with decreasing surface separation, the probabilities were normalised to the total reflected flux such that the integral at each $R_{\text{min}} - R_B$ is equal to 1. These distributions will be modulated by absorption at higher energies (above $E/V_B = 0.91$, below $R_{\text{min}} - R_B = 1.93$ fm (Supplementary Fig. 2a)) which will not be equally likely for all $E_x$. The data has been interpolated between measurement energies using Delaunay triangulation[57]. The black points show the mean excitation energy at each measured energy.

measuring the reflected (non-fused) flux at an energy well below the barrier, providing a snapshot of the system for a given minimum separation $R_{\text{min}}$ (see Fig. 1a, described in the Methods). This approach has previously been successfully applied for light nuclei[12,13]. By increasing the energy in small steps, we map out how the colliding nuclei evolve as they come closer together.

The reflected flux at each energy represents the integral of all reaction outcomes along a trajectory with a given $R_{\text{min}}$. The likelihood of transferring protons or neutrons increases exponentially with decreasing radial separation as the nuclear matter overlap increases. Thus the characteristics of the reflected flux should mainly represent processes occurring near the outer turning point of each trajectory. Critically, reactions that do lead to fusion must pass through the same

sequence of separations. Probing the characteristics of collisions not resulting in fusion thus probes the early stages of the fusion process.

We have studied $^{40}$Ca + $^{208}$Pb collisions, where systematics[10] indicate that above-barrier fusion cross-sections will be ~40% lower than model predictions. Previous measurements at above-barrier energies have indicated substantial probabilities of multinucleon transfer and large kinetic energy losses[14]. Being a collision of two spherical closed-shell nuclei, $^{40}$Ca + $^{208}$Pb provides a good benchmark for model development. Using the PRISMA magnetic spectrometer[15–17], we measured the distributions in mass ($A$), atomic number ($Z$) and kinetic energy of $^{40}$Ca + $^{208}$Pb reactions at 12 energies. We started at 20% below the fusion barrier, where there is negligible nuclear matter overlap (and fusion), increasing to 1% below the fusion barrier (see Methods). Measuring the kinetic energy as well as $A$ and $Z$ allowed us to reconstruct the excitation energy for each event, giving a complete characterisation of all the reflected nuclides.

## Results and discussion
### Rapidly increasing complexity
We find that there is a rapid change in the identities of the nuclei already outside the capture barrier, depending on $R_{\text{min}} - R_B$, the separation between the distance of closest approach $R_{\text{min}}$ and the barrier radius $R_B$. This is shown in Fig. 1b. The fraction of reflected flux $P/P_{\text{reflected}}$ remaining as $^{40}$Ca + $^{208}$Pb (lilac squares) is only $11.6 \pm 0.1\%$ at the closest separation distance measured $R_{\text{min}} - R_B = 0.46$ fm. Correlated with this, the smallest number of projectile-like and target-like nuclide pairs making up 95% of the reflected flux rapidly increases ($N_{95}$, green diamonds), reaching 31 different nuclide pairs. In Supplementary Fig. 1, we show the full distribution of the reflected flux in $N, Z$, and in Supplementary Fig. 2. the probability for 1–2 nucleon transfer and for multinucleon transfer. It is not just one or two channels contributing—there is a multitude of different mass and charge transfer processes occurring. In contrast, in a typical coupled-channels calculation[1] for this system, only states in $^{40}$Ca and $^{208}$Pb and a few simple transfer reactions would be approximately included.

Added to this complexity is the number of quantum states populated in the nuclei, revealed by the excitation energy $E_x$ distribution, shown in Fig. 1c. At large surface separations (low collision energies), the excitation energies are strongly peaked at $E_x = 0$ (the reflected nuclei are in their ground-states), but a tail extends to high $E_x$, becoming stronger as the nuclei approach closer ($R_{\text{min}} - R_B \lesssim 2.5$ fm). Here also the number of different nuclides produced rapidly increases. At the closest separation measured, the mean excitation energy (shown by the black circles) reaches $\langle E_x \rangle = 19.4$ MeV, where there a high density of (overlapping) quantum states in the interacting nuclei. The excitation energy is largest when multiple nucleons are transferred, with $\langle E_x \rangle = 29.5$ MeV (Supplementary Fig. 2c). Even the inelastic plus one & two nucleon transfer component shows a mean excitation energy of $\langle E_x \rangle = 10.0$ MeV (Supplementary Fig. 2b). Ground-state to ground-state transfers, as usually included in coupled-channels calculations, represent a negligible fraction of the reflected flux. Measuring at energies below the ($l$-dependent) barrier ensures that the probability of this reflected flux arising from capture is minimal. This is supported by the fact that signatures of capture are not yet present: the majority of the flux does not show mass flow towards symmetry nor are the mean excitation energies high enough for the kinetic energies to have been fully damped.

Significant amounts of multinucleon transfer products with high excitation energies have been previously observed at above-barrier energies[18–21], and seem to be a general feature of near-barrier heavy ion collisions. Significant energy loss (up to hundreds of MeV), associated with complex multinucleon transfers both towards and away from the target, is known as deep-inelastic scattering[14,22,23]. It has been identified as the energy loss mode in heavy-ion collisions[24] and has been modelled classically[22,25–27]. However, it has long been known[22] (also seen

here) that deep-inelastic scattering evolves smoothly from few-nucleon transfer and inelastic scattering, which must be treated quantum-mechanically[28].

How can we resolve this transition? In principle, when processes are fully reversible, coupled channels calculations will reproduce experiments if every coupling can be included. However, at high excitation energies where the density of states is very high, very many overlapping states will couple to each other in a complex scheme that results in a coupling that is effectively irreversible on the time scale of the nuclear collision (i.e. has a recurrence time longer than $10^{-21}$ s). This effective irreversibility leads to quasi-classical behaviour. An example of the scaling of recurrence times with system size in 1D superfluids is found in ref. 29.

How high does the excitation energy need to be to lead to (effective) irreversibility? As a concrete example, actinide nuclei having excitation energies larger than their fission barrier ($B_f \sim 6$ MeV) can fission (in $\lesssim 10^{-16}$ s) indicating thermalisation. For heavy systems we may thus expect that energy losses $\geq 6$ MeV (or perhaps lower) lead to (effectively) irreversible energy loss. A significant fraction of the reflected flux satisfies this condition. Crucially, we have shown that this energy loss begins outside the fusion barrier radius.

To briefly summarise: at the barrier radius, (i) the system consists of broad $Z, N$ distributions, with only a small probability of remaining as $^{40}$Ca and $^{208}$Pb nuclei and (ii) a significant fraction of collisions have high excitation energies consistent with (effective) energy dissipation. Neither condition is consistent with coherent coupled channels calculations, so how can they give even approximately correct results if they miss so much of the physics? We show that the answer lies in the correlation between $Z, N$ and $E_x$.

## Reconciling with barrier-passing models of fusion

We introduce a generalised variable that can quantify the effect of (multi-nucleon) transfer on fusion. A change in $Z$ before ($i$) and after ($f$) transfer results in a change in the Coulomb potential $V_i(R_{min}) - V_f(R_{min})$ at the distance of closest approach $R_{min}$. Additionally, transfer of nucleons (i.e changes in $Z$ and/or $N$) changes the nuclear binding energy, defined by the ground-state to ground-state Q-value $Q_{gg}$. The available energy for a given transfer (relative to the new potential) is thus

$$\Delta E_{gg} = Q_{gg} + (V_i(R_{min}) - V_f(R_{min})). \tag{1}$$

This energy may be in the form of kinetic or excitation energy ($E_x$). We can thus determine the kinetic energy with respect to the new potential at $R_{min}$ as:

$$\Delta E_{fi} = \Delta E_{gg} - E_x. \tag{2}$$

This is identical to $\Delta E_{fi} = (K_f - V_f(R_{min})) - (K_i - V_i(R_{min}))$, where $K_{i,f}$ are the total kinetic energies in the initial and final states. Thus, $\Delta E_{fi} > 0$ means that there is an increase in kinetic energy relative to the (new) potential, which increases fusion. $\Delta E_{fi} < 0$ decreases the kinetic energy relative to the new potential, resulting in reduced fusion. This idea is connected to what is partly incorporated in the semi-classical model GRAZING[30].

The $\Delta E_{gg}$ are shown by the red lines in Fig. 2a, b, with the height corresponding to the measured $P_{reflected}^{Z,N}$ for each measured $Z, N$. At $R_{min} - R_B = 3.87$ fm ($E/V_B = 0.80$) (Fig. 2a), the transfer probabilities are low, and $\Delta E_{gg}$ is strongly peaked at 0 MeV. At $R_{min} - R_B = 1.93$ fm ($E/V_B = 0.91$) (Fig. 2b) significant multinucleon transfer has begun, but there is little absorption by fusion to distort the overall distribution. Only 34% of the flux remains as $^{40}$Ca (seen at $\Delta E_{gg} = 0$), the rest being largely distributed between $\Delta E_{gg} = 0$ to 10 MeV, with a small fraction of events between 0 and $-15$ MeV (Fig. 2b, red lines). Since $\Delta E_{gg}$ is largely positive, for higher beam energies, one would expect enhanced fusion

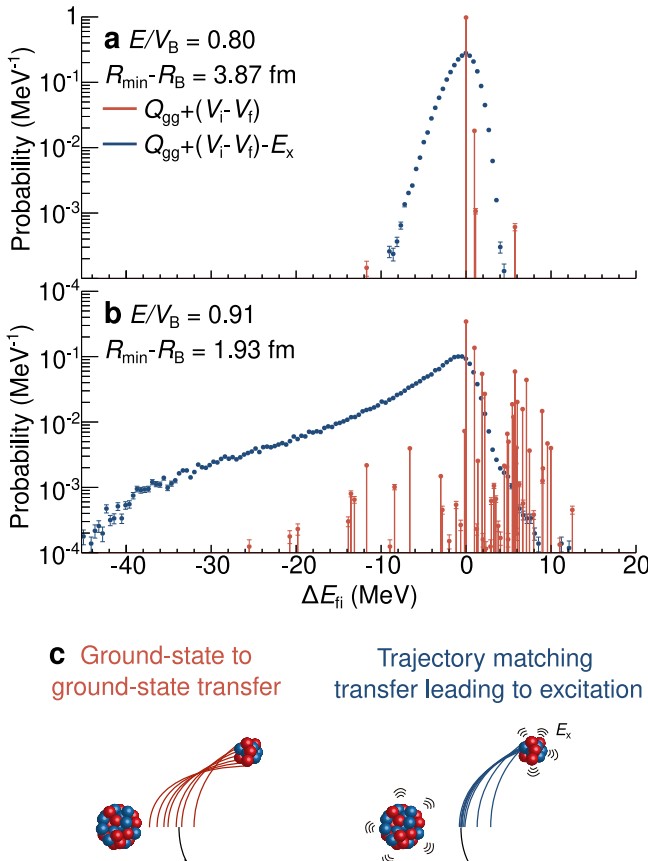

**Fig. 2 | Quantifying the change in available and kinetic energies.** The change in available energy $\Delta E_{gg}$ (red) and in kinetic energy $\Delta E_{fi}$ (blue) is shown for all transfer channels at (**a**) $E/V_B = 0.80$ ($R_{min} - R_B = 3.87$ fm) and (**b**) $E/V_B = 0.91$ ($R_{min} - R_B = 1.93$ fm). Cases for all other measured energies are shown in Supplementary Fig. 3. Statistical errors are shown. The red lines show $\Delta E_{gg}$, the maximum extra energy available to the colliding nuclei following transfer, which comprises kinetic and excitation energy. After subtracting the excitation energy, the blue curves show $\Delta E_{fi}$, the distribution of kinetic energies following transfer with respect to their potential. **c** Illustration showing how ground-state to ground-state transfers (red) result in discontinuities in the trajectories resulting from the change in potential after transfer. The reflected flux clustering to zero change in energy when the excitation energy is included (blue) is due to transfers preferentially producing excitation energies $E_x$ that ensure a smooth match between entrance and exit channels.

and at least a 10 MeV wide fusion barrier distribution[31] if nuclides were produced in their ground-states.

However, the nuclei are not produced in their ground states, but at a range of excitation energies. Taking $E_x$ into account event-by-event, the $\Delta E_{fi}$ distributions are shown by blue curves in Fig. 2, showing the actual change in kinetic energy relative to the new potential. Our determination of $E_x$ is critical to this interpretation. The many transfer channels with positive $\Delta E_{gg}$ at $E/V_B = 0.91$ now peak around $\Delta E_{fi} = 0$. Significantly, there is also an exponentially falling tail extending at least as far as $\Delta E_{fi} = -40$ MeV, that will reduce fusion at higher beam energies.

The peak at $\Delta E_{fi} = 0$ is due to favouring of continuous trajectories. Probabilities for transfer generally peak in a window around the Q-value ($Q = Q_{gg} - E_x$) that ensures that the linear and angular momenta in the entrance and exit mass partitions join smoothly, approximated by $Q + (V_i(R_{min}) - V_f(R_{min})) \approx 0$ MeV. This is known as the optimum Q-value[21,32–34] and is illustrated in Fig. 2c. Provided that the

optimum Q-value is $<Q_{gg}$, the equality can be satisfied if the fragments are excited, resulting in a peak at $\Delta E_{fi} = 0$. The tail for $\Delta E_{fi} < 0$ arises from (1) the exponential increase in the density of states with $E_x$ that will enhance transfer probabilities towards the high $E_x$ (low $\Delta E_{fi}$) side of the Q-window and (2) effectively irreversible multiple nucleon transfers in both directions (deep-inelastic scattering) that build up excitation energy[22,25]. While these measurements were made at a laboratory angle of 115°, the essential results are not expected to change if a different backwards angle (different $\ell$) was chosen[35,36], following corrections for the change in centrifugal energy[36].

### Consequences for superheavy element synthesis

Superheavy element synthesis requires that the captured nuclei evolve in shape to a compact equilibrated compound nucleus. The probability of doing so ($P_{CN}$) is very small because of strong competition from quasifission, in which the system re-separates into two heavy fragments before equilibration. The characteristics of quasifission, and by implication $P_{CN}$, depend most sensitively on the charge product $Z_1 Z_2$ of the colliding nuclei, but also on deformation, closed shells and matching of neutron to proton ratios of the colliding nuclei[37–41]. Our observations of the multitude of identities resulting from multinucleon transfer mean that each of these variables may be changed enroute to capture.

How could the fragmentation of flux observed here impact on superheavy element synthesis? Before capture, multinucleon transfer results in a distribution of $Z_1 Z_2$. Shown in Fig. 3a is the $Z_1 Z_2$ distribution for collisions with $|\Delta E_{fi}| < 5$ MeV (those having similar probabilities of capture to the starting value) at $E/V_B = 0.91$ ($R_{min} - R_B = 1.93$ fm). The charge product distribution has a tail to much lower $Z_1 Z_2$. For the lower $Z_1 Z_2$ collisions (shown for $^{36}S + ^{208}Pb$ in Fig. 3b) the fragments show narrow fission-like mass distributions having no correlation with angle.

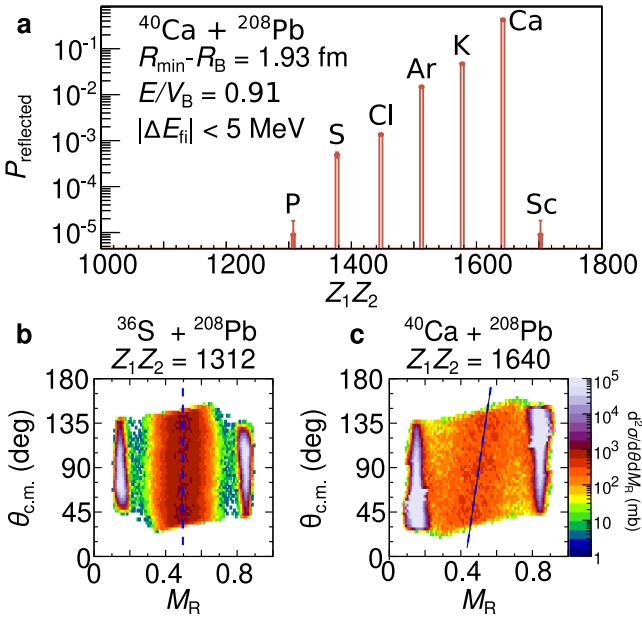

**Fig. 3 | Potential impact of transfer on superheavy element formation. a** The distribution of charge products between the projectile-like and target-like nuclides $Z_1 Z_2$ at the below-barrier energy of $E/V_B = 0.91$, with corresponding to a separation of $R_{min} - R_B = 1.93$ fm. Shown are the reflected nuclei that maintain similar kinetic energies with respect to the barrier, i.e. $|\Delta E_{fi}| < 5$. Other energies are shown in Supplementary Fig. 4. The y-axis shows the absolute probability of the flux being reflected and having charge product $Z_1 Z_2$, $P_{reflected}(Z_1 Z_2)$. Fission-like fragment mass-ratio $M_R$ vs scattering angle $\theta_{cm}$ distributions for (**b**) $^{36}S + ^{208}Pb$ ($E/V_B = 1.067$) and (**c**) $^{40}Ca + ^{208}Pb$ ($E/V_B = 1.058$). The intense vertical bands at the extremes of the $M_R$ distributions arise from (quasi)elastic scattering. The blue dashed lines guide the eye, indicating the degree of mass-angle correlation and thus sticking times.

This indicates many rotations and long sticking times, associated with larger $P_{CN}$. In contrast for $^{40}Ca + ^{208}Pb$ with higher $Z_1 Z_2$ (Fig. 3c), there is a strong correlation of fragment mass with angle, showing that the system typically comes apart in less than half a rotation. This is associated with smaller $P_{CN}$.

For reactions forming superheavy elements with much larger $Z_1 Z_2$, and $P_{CN}$ values perhaps as low as $10^{-6}$, a much stronger dependence of $P_{CN}$ on $Z_1 Z_2$ would be expected, making the larger $P_{CN}$ for lower $Z_1 Z_2$ significantly more impactful. We thus speculate that the multinucleon transfer processes occurring outside the capture barrier radius may provide a mechanism for superheavy element synthesis. Multinucleon transfer yields and $N, Z$ distributions depend strongly on the colliding system[42–45], and this may also explain the observed isotopic difference in fusion probabilities[40]. These ideas need to be tested quantitatively through further experimental measurements, in particular for deformed actinide nuclei, to see whether the characteristics agree with the present measurements with closed-shell spherical nuclei.

In summary, in collisions of $^{40}Ca$ and $^{208}Pb$ nuclei, we find that the nuclei reach the fusion barrier radius with a multitude of proton and neutron numbers, having broad distributions of excitation energies reaching tens of MeV. In contrast, standard models of fusion have assumed that nuclei reach the point of capture essentially unchanged– in just a handful of low-lying states.

The effect of these distributions on fusion can be seen through a variable we introduce, $\Delta E_{fi}$, showing how the energies with respect to the barrier change after multinucleon transfer. Due to transfer favouring smooth trajectories, most events have a similar energy with respect to their barrier, and thus a similar fusion probability to that of the initial $^{40}Ca$ and $^{208}Pb$ collision. This explains why standard models of fusion work as well as they do, despite missing the major physical processes occurring in the dynamics. We attribute the observed correlation to the favouring of smooth trajectories before and after transfer.

Importantly, we observe a significant tail of events having much higher excitation energies (negative $\Delta E_{fi}$) and thus lower kinetic energies. These will reduce fusion, explaining long-standing experimental observations[6,10]. The distribution of $\Delta E_{fi}$ is the reason that fusion barrier distributions for higher $Z_1 Z_2$ reactions are smoothed and have a tail extending to high energies. Such a barrier distribution is seen in $^{20}Ne + ^{208}Pb$[35], which cannot be reproduced in a reasonably constrained coupled-channels calculation even when a very large number of states are included[46].

Our results thus explain both why standard models of fusion seem to work at all, and also why they fail. Our results offer a framework to develop more realistic models of nuclear fusion, including the processes actually occurring in the early stages of the pathway to fusion.

## Methods

### Reflected flux

**Experimental details.** The measurements of the reflected flux were performed at Legnaro National Laboratory XTU Tandem-ALPI accelerator complex, using the PRISMA magnetic spectrometer[15–17]. PRISMA features a large solid angle (80 msr, $\Delta\theta_{lab} = \pm 6°$, $\Delta\phi = \pm 11°$), momentum acceptance $\Delta p = \pm 10\%$, mass-resolution $\Delta A/A \sim 1/200$, and energy resolution up to 1/1000 (via time-of-flight measurement). In this experiment, PRISMA was located at $\theta_{lab} = 115°$.

The magnetic fields were set for each energy to maximise the transmission for the dominant charge state of the elastically scattered beam. Thus, the measurements focus on the evolution of quasielastic scattering to multinucleon transfer (or deep-inelastic scattering). The finite momentum acceptance of PRISMA means that binary reaction channels with $\Delta p > \pm 10\%$–those with much larger changes in $N, Z$–cannot be observed. In particular, the expected smooth evolution from multinucleon transfer to quasifission[38,47,48] will not be observed. This makes our near-barrier measurements

lower limits for the extent of multinucleon transfer in $Z$, $N$, and thus also energy dissipation. Additionally, if present, sticking and rotation in the multinucleon transfer component may mean that the measurement at $\theta_{lab} = 115°$ may be contaminated by trajectories originating from smaller angular momenta, closer to their effective barrier. This is difficult to quantify without knowledge of the sticking times, which can vary widely depending on the model interpretation. However, we note that the onset of multinucleon transfer at $E/V_B \sim 0.88$ (for $\theta_{lab} = 115°$) corresponds to $E/V_B = 0.94$ for $\ell = 0$, still well below the barrier.

Beams of $^{40}$Ca were produced in 12 energy steps between $E_{cm} = 189.0$ and 230.5 MeV. For the energies above 213 MeV, where the ALPI booster accelerator was used, carbon degrader foils of 135 μg/cm$^2$ or 205 μg/cm$^2$ were employed to provide three beam energies for each accelerator tune. The $^{40}$Ca beams were delivered onto ~150 μg/cm$^2$ $^{208}$PbS targets oriented with their normals at 60° to the beam axis. The targets had 20 μg/cm$^2$ carbon backings which were placed upstream of the target such that the particles accepted into PRISMA did not pass through the carbon backing.

**Data analysis.** Absolute probabilities of the integrated reflected flux $P_{reflected}$ were determined by normalising to Rutherford scattering yields in two silicon beam monitoring detectors placed at forward angles on either side of the beam axis (see Supplementary Fig. 2a). To account for the transmission through PRISMA, it was assumed that at the lowest energy ($E/V_B = 0.80$), $P_{reflected} = d\sigma_{reflected}/d\sigma_{Rutherford} = 1$. The efficiency of PRISMA is rather flat, except for at the edges of its acceptance[49]. Therefore, the overall shape of the measured distributions are not substantially moderated by acceptance effects, thus allowing qualitative comparisons of their evolution with energy.

The atomic (proton) number $Z$, mass number $A$ and energies of the scattered beam-like particles passing through PRISMA were determined using the $TOF - B\rho - \Delta E$ technique[15]. Ions pass through a position-sensitive microchannel plate timing detector (MCP)[50] before entering the quadrupole and dipole magnets. At the focal plane, ions first pass through a multi-wire parallel plate avalanche counter (MWPPAC) then into a segmented ionisation chamber[51]. The measured positions of the ions in the MCP and MWPPAC define the trajectory of the ions through the magnetic elements, determining the magnetic rigidity $B\rho$. The energy loss of ions in the ionisation chamber enables the determination of $Z$, and with $B\rho$, the charge-state $q$. Together with time-of-flight (TOF), this allows determination of $A$ (and hence neutron number $N = A - Z$) and the kinetic energy of the projectile-like nuclei. The resulting $(Z, N)$ distributions are shown in Supplementary Fig. 1.

Following $Z$, $N$ determination, the ground-state to ground-state energy difference (Q-value, $Q_{gg}$) could be obtained for each event. With the kinetic energy information, the total excitation energy $E_x$ could be derived ($E_x = Q_{gg} - Q$), making use of two-body kinematics. Crucially, determining $Z$, $N$ and $E_x$ rather than total kinetic energy loss (TKEL) allowed us to calculate the change in energy with respect to the barrier after transfer relative to that of the entrance channel. This allowed a direct link to fusion hindrance to be made (Supplementary Figs. 2b, c, 3).

**Determination of distance of closest approach.** Mapping from a beam energy to a distance of closest approach requires that we determine the point at which the incoming kinetic energy is matched by the potential, as illustrated in Fig. 1 of the main text. We begin by constructing the total inter-nuclear potential $V_{tot}$, being the sum of an attractive nuclear potential $V_{nuc}$, and the repulsive Coulomb $V_{coul}$ and centrifugal $V_{cent}$ potentials, $V_{tot} = V_{nuc} + V_{coul} + V_{cent}$. $V_{nuc}$ was calculated using the São Paulo potential[52], a density dependent double-folding potential, with an energy dependent correction arising from

Pauli non-locality, determined from heavy-ion scattering data. This potential has no free parameters.

$V_{coul}$ is the repulsive Coulomb potential between a positively charged finite sphere and a positive-point charge, where the radius of the sphere (the Coulomb radius) is determined using São Paulo systematics[52].

$V_{cent}$ is the effective centrifugal potential depending on the angular momentum $l$, $V_{cent} = \frac{\hbar^2 l(l+1)}{2\mu R^2} = \frac{L^2}{2\mu R^2}$, where μ is the reduced mass of the colliding nuclei. Since the measurements of the reflected flux were performed at a fixed laboratory angle $\theta_{lab} = 115°$, an increase in energy $E_{cm}$ corresponds to a small decrease in $l$ (and thus a small decrease in the centrifugal potential) for particles scattered to $\theta_{lab}$.

We thus determine the angular momentum for particles scattered to 115° at each energy, $l_{E_{cm}, \theta = 115°}$, assuming Rutherford trajectories. In Rutherford scattering, $L$ is related to the impact parameter $b$ via $L = \mu v_0 b$, where $v_0 = \sqrt{2E_{cm}/\mu}$, and $b = \frac{Z_1 Z_2 e^2}{2 E_{cm} \tan \theta/2}$.

Once $V_{tot}$ have been constructed for each $l_{E_{cm}, \theta = 115°}$, the distance of closest approach between centers $R_{min}$ at each $E_{cm}$ at $\theta_{lab} = 115°$ was determined by solving to find the outside intersection of $E_{cm}$ and $V_{tot}$. $R_B$ and $V_B$ are found as the local maxima of $V_{tot}$. The inter-nuclear potentials as a function of $R_{min} - R_B$ are shown by the blue curves in Fig. 1 of the main text, and the summed $V_{cent} + V_{coul}$ in green. The distances of closest approach for each energy indicated by the yellow crosses, and these, and the energies with respect to the ($l_{E_{cm}, \theta = 115°}$) barrier are tabulated in Supplementary Table 1.

## Fission and quasifission mass distributions

The measurements of the fission and quasifission mass distributions were performed at the Heavy Ion Accelerator Facility, located at the Australian National University, Canberra, Australia. Beams of $^{36}$S, $^{40}$Ca were delivered by the 14UD 15 MV electrostatic pelletron accelerator. The beams were delivered to a $^{208}$PbS targets ranging in thickness from 100 to 170 μg/cm$^2$. The targets were placed with their normals oriented at 60° to the beam axis to minimise energy loss of the fission fragments in the targets, and avoid shadowing of the detectors by the target frame. In order to compare the fission mass distributions across the two different systems, the energies were chosen to be between 6% and 7% above the fusion barrier[53] for each system.

Fission and quasi-fission fragments were detected in coincidence using the CUBE spectrometer, in this experiment consisting of two multiwire proportional counters (MWPCs) with active areas of $279 × 357$ mm$^2$. The MWPCs were placed 180 mm from the target, with one detector at backwards angles centered at 90° continuously covering 55° to 130° and the other at forward angles, centered at 45°, covering 5° to 80°[54]. The typical azimuthal coverage was 70°.

Position information $(\theta, \phi)$ was extracted from the $X$ and $Y$ anode planes of each MWPC, comprising grids of 20 μm gold-plated tungsten wires with 1 mm spacing. The central 0.9 μm gold-coated mylar cathode provided the timing information. From the position and timing information, the fission fragment velocities, energies and mass ratios $(M_R)$ were determined in the center-of-mass frame using energy-momentum conservation[37,55]. Fission fragment source analysis, confirming the fission fragments as being binary events originating with the $^{208}$Pb in the targets was performed. This is done by selecting the events where the components of the fission fragment velocities in the perpendicular $v_\perp$ and parallel $v_\parallel$ directions relative to the beam are consistent with full-momentum transfer fission after reactions with $^{208}$Pb. That is, the events are tightly centered around $v_\perp = 0$ and $[v_\parallel - v_{CN}] \sin \theta_{cm} = 0$, where $v_{CN}$ is the velocity of the compound nucleus, and $\sin \theta_{cm}$ the scattering angle[56].

Two silicon monitor detectors were placed at laboratory angles of $\theta = 30°$ and $\phi = 90°, 270°$ to measure elastically scattered events for absolute cross-section determination $d\sigma^2/dM_R d\theta$.

## Data availability

The data generated in this study have been deposited in the Australian National University Data Commons and is available at https://doi.org/10.25911/zkq5-7187.

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

## Acknowledgements

This work was supported by Australian Research Council Grants DP190101442 (E.C.S., M.D.), DP200100601 (D.J.H., M.D.), DE230100197 (K.J.C.), DP230101028 (D.J.H., M.D.). T.M. and S.S. acknowledge the support of the Croatian Science Foundation under Project no. 7194 and Project no. IP-2018-01-1257. Support for Heavy Ion Accelerator Facility operations through the NCRIS program is acknowledged. The efforts of the accelerator staff at the INFN Legnaro National Laboratory and at the Heavy Ion Accelerator Facility are gratefully acknowledged.

## Author contributions

K.J.C. performed the high-level analysis, interpreted the results, and wrote the paper. D.C.R. performed the initial data analysis and interpretation of the data collected with PRISMA. D.J.H. and M.D. conceived of the experiments and supervised the analysis and interpretation of the data and the manuscript preparation. E.C.S. provided key theoretical insights. D.Y.J. performed the data analysis of the fission-like mass-angle distributions. M.E. was the spokesperson of the PRISMA experiment. The PRISMA collaboration, L.C., E.F., T.M., G.M., A.M.S, and S.S. built and characterised the PRISMA spectrometer and analysis techniques. L.C., E.F., T.M., G.M., A.M.S., S.S., D.H.L., and E.M. ran the PRISMA experiment and gathered these data. D.J.H., M.D., M.E., D.H.L. and D.C.R. ran the experiments with the CUBE, spectrometer and gathered these data, with extensive support from N.L., D.J.H. built and characterised the CUBE spectrometer as well as conceiving the analysis methods.

## Competing interests

The authors declare no competing interests.
