## [Peer Review File · Nature Communications]

Colliding heavy nuclei take multiple identities on the path to fusionReviewers' comments:

Reviewer #1 (Remarks to the Author):

Dear authors;

Report of the Referee- Colliding heavy nuclei take multiple identities on the path to fusion/K.J. Cook et al.

It is an extremely difficult problem to understand and elucidate the location of multinucleon rearrangement in low energy heavy ion nuclear reactions. In the present work, by measuring the reflected

flux for $^{40}\text{Ca}+^{208}\text{Pb}$ reaction system at an energy well below the barrier can provide a snapshot of

the system for a given minimum separation. The authors find that the colliding nuclei take many different identities in the early stages of collisions (before capture), the measured results show an 'explosion' of

mass and charge transfers between the nuclei (over 90 nuclide pairs) before capture occurs, with much higher probability and complexity than expected.

Therefore, the mechanism of superheavy nucleus production may be more complex than previously expected. The current study provides a relatively novel idea for understanding low-energy nuclear reactions.

In principle, it is difficult to identify whether the reaction of interest is mainly controlled by the nucleus-nucleus potential or

by the mode of energy dissipation.

The present work will be a guide to future theoretical development.

So, I recommend this paper for publication in Nature Communication after some improvements.

1. How do the authors distinguish experimentally whether the yield of reflected flux comes from before or after capture. The author should try to explain the physical reason or criterion.

2. The correlation between Z, N and E_x is significantly dependent on angular momentum. The authors should discuss the influence of angular momentum on results and conclusions.

Reviewer #2 (Remarks to the Author):

Fusion reactions constitute one of the most important aspects of nuclear experiment and theory and is vital for other fields such as astrophysics. The prevailing problem in the theoretical understanding of these reactions is the lack of a fundamental approach to address the many-body tunneling process. Due to this the problem is reduced to an effective barrier, either phenomenological or models that rely on the time-dependent density functional theory (TDDFT). The phenomenological models mostly assume that the nuclei do not change their character as they begin to overlap and other effects are included as an afterthought, such as particle transfer, neck formation, and nuclear excitations. Their success mainly stems from adjustable parameters whose origin is not well understood. The TDDFT approach does take into account the dynamical changes during the reaction at above barrier energies at the mean-field level but cannot directly explain the tunneling process due to its semi-classical nature. Furthermore, most theories assume a single reaction outcome during the collision process, which is quantum mechanically not correct for most reactions, particularly for sub-barrier fusion.

The authors are performing novel experiments to elucidate the early stages of fusion and its impact on the effective potential barrier. This is fundamentally important and could help improve our understanding of the evolution of the fusion process and aid the development of new theoretical approaches. The authors propose to probe the reaction process by studying the reflected flux to clarify the events that would not be included in theoretical models.

Authors find a number of very important results that challenges almost all theoretical approaches. First, is the number of participating quantum states due to the excitation energy content of the interacting system They point out that standard inclusion of excitations in coupled-channels represent a negligible fraction of these states.

In the next section the authors try to explain the success of the coupled-channels approach despite the above mentioned disparities. While I follow the arguments made here, they are somewhat heuristic. First, after charge exchange R_{\min} value may also change but more importantly the nuclei are dynamically changing and the E_{QQ} values may not make sense for overlapping systems, i.e. the excitations are not with respect to ground states but the states of the combined system (see for example PRC80, 041601 (2009)).

Theoretical models make other approximations as well, such as the determination of the inner turning point at the barrier, which had to be corrected by the shallow potential approach phenomenologically. Also, approximations related to tunneling methods used. The deep barrier tunneling of two nuclei is a very complex problem that even the best theoretical minds have not been able to explain microscopically over the years. So, all models should be taken with a grain of salt.

The authors also use their arguments to point out why the multinucleon reactions may provide a new avenue in the search for new superheavy elements, which has been speculated theoretically.

In summary, I think the paper provides a very important contribution to the understanding of nuclear fusion and particularly points out the deficiencies in the current theoretical approaches and provides guidance for future theories. I

recommend publication and provide the following comments that should be considered.

OTHER COMMENTS:

1. In the fourth paragraph "turning point" should be "outer turning point".
2. Fig. 1(a) should say something like "depicted is the..." since this is still derived from the reduction of the interaction energy using a model.
3. Fig. 1(a) is not mentioned in text explicitly.
4. At the end of paragraph 1 of Sec. 2.2.1 "would be included" should say "would be approximately included" since this inclusion is done via the modeling of nuclear excitations using vibrational/rotational models of the entrance channel nuclei.
5. Fig.1(c) caption, the sentence before the last ends with "likely at all E_x ", rather use "for" instead of "at".
6. The nuclei considered here are closed shell spherical nuclei. Naturally, if one or both of these nuclei were deformed some of these arguments will have to be modified, specially the superheavy element formation where the target is usually an actinide.
7. Fig.2's a,b,c's should be in parentheses and larger font and the next sentence after the figure, Fig.2 should be Fig.2(a,b)

Reviewer #3 (Remarks to the Author):

The authors studied multinucleon transfer (MNT) reactions of the $Ca40+Pb208$ system at energies below the Coulomb barrier using the PRISMA magnetic spectrometer. Nuclear charge and mass of ejectile nuclei were identified in event-by-event basis. Also, using the reaction Q-value and kinetic energy of ejectile nucleus, authors derived excitation energy of the system at the MNT reaction. The authors systematically changed incident beam energy, which is correlated to the minimum distance that the colliding nuclei face with each other, in order to investigate the change of number of transferred neutrons/protons, as well as average excitation energy and excitation-energy distribution. As the incident energy increases, number of opened transfer channels and excitation-energy increases. The authors proposed an expression to represent the "available energy" for a given transfer process,

calculated by the change of ground-to-ground state Q-value and the change of Coulomb energy determined by identifying the outgoing two nuclides. They define that the available energy works to change the fusion reaction, due to the redistribution of the Coulomb barrier. Considering the loss of the available energy by the excitation of the system, the kinetic energy with respect to the new potential, working as the driving energy for fusion, is discussed.

The experiment is well organized, and they obtained an interesting new data with high quality and reliability. The data should be published in some journal.

However, I see a serious problem in their interpretation. They argue the impact of the MNT process on fusion. The idea that the change of the Coulomb barrier after the MNT influence the subsequent fusion is a strange argument. To realize this, fusion must happen after the MNT process finishes. For the case of incomplete fusion, break up of weakly bound incident nucleus can happen, and simultaneously one of the clusters can fuse to target nucleus by emitting the other part of nucleus. On the other hand, in this experiment, they are looking only ejectile nuclei in the vicinity of Ca40. There might be the case that some nucleons (cluster) can go fusion after the MNT reaction, but this was not identified in the experiment by measuring evaporation residues in coincidence. Even it could happen, a nucleus slightly large than Pb208 target is formed. I question if such a channel is important for fusion, especially for the discussion trying to understand the synthesis of SHEs.

In more generally, if the MNT reactions happens before fusion, many kinds of compound nuclei are produced. If we follow the measurement and argument by the authors, the number of nuclear species as compound nuclei produced after the MNT reaction needs to be expanded significantly already at the incident energy at the Coulomb barrier. So far, however, there is no such an experimental evidence in the measurement of evaporation residues using in-flight kinematic separator. Even at the beam energies higher than the Coulomb barrier, the produced evaporation residues can be interpreted only assuming one specific compound nucleus species, represented by the sum of protons and neutrons for projectile and target nuclei, and different evaporation channels.

I find the authors have a serious misinterpretation in the reaction, and the argument is non-logical. Thus, I do not agree this article to be published in Nature Communications. What I can suggest is that the authors delete the discussion on the links to fusion process, and simply discuss the experimental results of MNT reaction. After the manuscript is revised, the authors would submit it in another journal devoted to nuclear physics instead of Nature Communications, as the experiment was based on the same process already established and similar experiments had been repeatedly carried out using the same devices.

For future consideration, some of the sentence/expression is ambiguously written, so it is hard to understand even for researchers in this field. This must be improved.

L36 "Superheavy elements exist at the limits of both physics and chemistry"

- The sentence looks strange to me.

L41 "models overestimate experimental capture and fusion cross-sections."

- I can imagine there must be a model which underestimate experimental data. Why only models overestimating the data are discussed here.

L42 "explosion"

- What does this word means to represent collision between heavy nuclei.

L47 "Our interpretation overturns the current picture of fusion"

- The sentence is too strong. I do not agree the investigation in this article overturns the current understanding on fusion.

L66 "the identities of the nuclei are essentially unchanged prior to this point."

- Concerning "the identifies of the nuclei", what kind of properties of nucleus and the impacts on fusion reactions the authors want to argue?

L117 "the smallest number of nuclide pairs making up 95% of the reflected flux rapidly increases (N95, green diamonds), reaching 31 distinct nuclide pairs."

- What does the pairs mean ? One pair require two nucleons. So 31 pairs means the exchange of 62 nucleons ?

L121 "It is not just one or two channels contributing –there is a multitude of different mass and charge transfer processes occurring. In contrast, in a typical coupled-channels calculation [1] for this system, only states in ^{40}Ca and ^{208}Pb and a few simple transfer reactions would be included."

- I guess to explain the current experimental data of $\text{Ca}^{40}+\text{Pb}^{208}$ MNT reaction is already outside the scope in the model of [1]. So, this sentence can be deleted.

L134 "The excitation energy is largest when multiple nucleons are transferred, with $\langle E_x \rangle = 29.5$ MeV, though even the inelastic and one & two nucleon transfer component shows a mean excitation energy of $\langle E_x \rangle = 10.0$ MeV"

- In Fig.1(c), there is no data point showing average excitation energy $\langle E_x \rangle = 29.5$ MeV. What does it mean ?

L177 "Significant energy loss (up to hundreds of MeV), associated with complex multinucleon transfers in both directions, is known as deep-inelastic scattering [14, 24, 25]."

- What does "in both directions" mean ?

L194 "How high does the excitation energy need to be?"

- This sentence is strange.

L270 "one would expect enhanced fusion and at least a 10 MeV wide fusion barrier distribution."

- I do not agree this argument. The discussion is only for the MNT process. How can we argue if the remaining nucleons after ejecting Ca40-like nuclei can go fusion ?

L279 "Significantly, there is also an exponentially falling tail extending at least as far as $\Delta E_{fi} = -40$ MeV, that will reduce fusion."

- Again from this experimental data, one cannot say anything on fusion process.

L330 "Before capture, multinucleon transfer results in a distribution of Z1Z2."

- Again there is no experimental evidence that capture process happens after multinucleon transfer (MNT) process finishes. The authors measured only MNT ejectiles and nothing is measured to argue the evidence of subsequent fusion.

L327 "Our observations of the multitude of identities mean that each of these variables may be changed enroute to capture."

- Again it is hard to believe fusion and quasifission processes are influenced by the MNT reaction. They are competing and independent process. Two-step process involving the MNT and fusion/quasifission has not been identified experimentally. What can be changed is only their cross sections due to the loss of incident flux by the MNT process.

Fig.3 The authors show the distribution of the yield of ejectile nuclei. This is the results of MNT reaction, and this does not mean fusion/quasifission can happen from these different Z1Z2 combinations.

L582 “The overall shape of the distributions will not be substantially moderated by the acceptance of PRISMA, except at the edges of the acceptance, reached only at the higher beam energies, thus allowing qualitative comparisons of their evolution with energy.”

- The meaning is not clear to me.

Fig.4 (caption) “Diagonal lines indicate the isospin asymmetry equal to that of the compound nucleus ^{248}Fm ($N/Z = 1.45$)”

- N/Z for ^{248}Fm is 1.48 (not 1.45).

Response to referees

We thank the reviewers for their comments on our manuscript, and address each comment individually. We have shown referee comments *in italics* and our responses are indented in normal text.

Reviewer #1 (Remarks to the Author):

Dear authors;

Report of the Referee- Colliding heavy nuclei take multiple identities on the path to fusion/K.J. Cook et al.

It is an extremely difficult problem to understand and elucidate the location of multinucleon rearrangement in low energy heavy ion nuclear reactions. In the present work, by measuring the reflected flux for $^{40}\text{Ca}+^{208}\text{Pb}$ reaction system at an energy well below the barrier can provide a snapshot of the system for a given minimum separation. The authors find that the colliding nuclei take many different identities in the early stages of collisions (before capture), the measured results show an ‘explosion’ of mass and charge transfers between the nuclei (over 90 nuclide pairs) before capture occurs, with much higher probability and complexity than expected. Therefore, the mechanism of superheavy nucleus production may be more complex than previously expected. The current study provides a relatively novel idea for understanding low-energy nuclear reactions. In principle, it is difficult to identify whether the reaction of interest is mainly controlled by the nucleus-nucleus potential or by the mode of energy dissipation. The present work will be a guide to future theoretical development. So, I recommend this paper for publication in Nature Communication after some improvements.

1. How do the authors distinguish experimentally whether the yield of reflected flux comes from before or after capture. The author should try to explain the physical reason or criterion.

This is an important clarification! By making measurements at below barrier energies, we are minimising the capture probability. By doing so, we expect that the yield of the reflected flux arises primarily from interactions prior to capture. The experimental data supports this – re-separation after capture (e.g quasifission) gives fragments that have experienced mass-flow towards symmetry and full kinetic energy damping. Our measurements show that the majority of the flux has exchanged nucleons that move towards (but doesn’t achieve, on average) equilibration in N/Z (expected to occur first [Simenel, Godbey and Umar, PRL 212504 (2020)]) and with high excitation energy, but not full damping.

In order to make this point in the paper, we have added the following to the end of the second paragraph in section 2.21:

“Measuring at energies below the (*l*-dependent) barrier ensures that the probability of this reflected flux arising from capture is minimal. This is supported by the fact that signatures of capture are not yet present: the majority of the flux does not show mass flow towards symmetry nor are the mean excitation energies are high enough for the kinetic energies to be fully damped.”

2. The correlation between Z,N and Ex is significantly dependent on angular momentum. The authors should discuss the influence of angular momentum on results and conclusions.

Thank you for this excellent suggestion. A priori, one might expect this to be significant. However, measurements of the quasielastic events at a range of backward angles used to

determine barrier distributions [Piasecki PRC 85 054608 (2012), Timmers Nucl. Phys. A. 584 (1995) 190-204] show that the “available energy” with respect to the barrier (defined in our manuscript) is the key quantity impacting fusion, and it appears to be largely independent of angle (thus l) after correcting for the centrifugal energy [Timmers Nucl. Phys. A. 584 (1995) 190-204]. Then, at energies around the barrier, changes of the *overall* distributions and correlations between Z, N and E_x can be taken care of by a centrifugal energy correction (while of course the details may change).

To address this in the manuscript, we have added the following to the end of section 2.2.2, where we discuss the distribution of available energies:

“While these measurements were made at a laboratory angle of 115° , the essential results are not expected to change if a different backwards angle (different θ_{lab}) was chosen [cite{Piasecki2012,Timmers1995}], following corrections for the change in centrifugal energy [cite{Timmers1995}].”

Reviewer #2 (Remarks to the Author):

Fusion reactions constitute one of the most important aspect of nuclear experiment and theory and is vital for other fields such as astrophysics. The prevailing problem in the theoretical understanding of these reactions is the lack of a fundamental approach to address the many-body tunneling process. Due to this the problem is reduced to an effective barrier, either phenomenological or models that rely on the time-dependent density functional theory (TDDFT). The phenomenological models mostly assume that the nuclei do not change their character as they begin to overlap and other effects are included as an afterthought, such as particle transfer, neck formation, and nuclear excitations. Their success mainly stems from adjustable parameters whose origin is not well understood. The TDDFT approach does take into account the dynamical changes during the reaction at above barrier energies at the mean-field level but cannot directly explain the tunneling process due to its semi-classical nature. Furthermore, most theories assume a single reaction outcome during the collision process, which is quantum mechanically not correct for most reactions, particularly for sub-barrier fusion.

The authors are performing novel experiments to elucidate the early stages of fusion and its impact on the effective potential barrier. This is fundamentally important and could help improve our understanding of the evolution of the fusion process and aid the development of new theoretical approaches. The authors propose to probe the reaction process by studying the reflected flux to clarify the events that would not be included in theoretical models.

Authors find a number of very important results that challenges almost all theoretical approaches. First, is the number of participating quantum states due to the excitation energy content of the interacting system They point out that standard inclusion of excitations in coupled-channels represent a negligible fraction of these states.

In the next section the authors try to explain the success of the coupled-channels approach despite

the above mentioned disparities. While I follow the arguments made here, they are somewhat heuristic. First, after charge exchange R_{\min} value may also change but more importantly the nuclei are dynamically changing and the E_{QQ} values may not make sense for overlapping systems, i.e. the excitations are not with respect to ground states but the states of the combined system (see for example PRC80, 041601 (2009)). Theoretical models make other approximations as well, such as the determination of the inner turning point at the barrier, which had to be corrected by the shallow potential approach phenomenologically. Also, approximations related to tunneling methods used. The deep barrier tunneling of two nuclei is a very complex problem that even the best theoretical minds have not been able to explain microscopically over the years. So, all models should be taken with a grain of salt.

The authors also use their arguments to point out why the multinucleon reactions may provide a new avenue in the search for new superheavy elements, which has been speculated theoretically.

In summary, I think the paper provides a very important contribution to the understanding of nuclear fusion and particularly points out the deficiencies in the current theoretical approaches and provides guidance for future theories. I recommend publication and provide the following comments that should be considered.

OTHER COMMENTS:

1. In the fourth paragraph "turning point" should be "outer turning point".

This is an important clarification -- we have corrected this to be "should mainly represent processes occurring near the outer turning point of each trajectory"

2. Fig. 1(a) should say something like "depicted is the..." since this is still derived from the reduction of the interaction energy using a model.

Thank you, this is a good point. We have changed the caption to start with "Depicted is the internuclear potential..."

3. Fig. 1(a) is not mentioned in text explicitly.

It was already mentioned explicitly in the first and third paragraphs of the introduction, so we have not added more in-text references.

4. At the end of paragraph 1 of Sec. 2.2.1 "would be included" should say "would be approximately included" since this inclusion is done via the modeling of nuclear excitations using vibrational/rotational models of the entrance channel nuclei.

Thank you for this suggestion, we have changed the final sentence to read "simple transfer reactions would be approximately included." as suggested.

5. Fig.1(c) caption, the sentence before the last ends with "likely at all E_x ", rather use "for" instead of "at".

We have made this change.

6. *The nuclei considered here are closed shell spherical nuclei. Naturally, if one or both of these nuclei were deformed some of these arguments will have to be modified, specially the superheavy element formation where the target is usually an actinide.*

Indeed! We might expect that there is a significant variation in the multi-nucleon transfer yields going from below the barrier (where you would see tip-driven dynamics) to above-barrier, as side collisions become possible. TDHF and TDRPA calculations [Simenel, Godbey and Umar, PRL 212504 (2020)] show that the timescales for multinucleon transfer and kinetic energy damping are all very similar for symmetric, closed shell and superheavy element synthesis calculations. Here, we focused on closed-shell spherical nuclei as an initial study, but extending to more deformed systems would be a very interesting avenue for future research. We have modified the last line of section 2.2.3 to read

“These ideas need to be tested quantitatively through further experimental measurements, in particular for deformed actinide nuclei, to see whether the characteristics agree with the present results for closed-shell spherical nuclei.”

7. Fig.2's a,b,c's should be in parentheses and larger font and the next sentence after the figure, Fig.2 should be Fig.2(a,b)

We have made these two changes.

Reviewer #3 (Remarks to the Author):

[In this case, we have highlighted in blue the text we disagree with.]

The authors studied multinucleon transfer (MNT) reactions of the Ca40+Pb208 system at energies below the Coulomb barrier using the PRISMA magnetic spectrometer. Nuclear charge and mass of ejectile nuclei were identified in event-by-event basis. Also, using the reaction Q-value and kinetic energy of electile nucleus, authors derived excitation energy of the system at the MNT reaction. The authors systematically changed incident beam energy, which is correlated to the minimum distance that the colliding nuclei face with each other, in order to investigate the change of number of transferred neutrons/protons, as well as average excitation energy and excitation-energy distribution. As the incident energy increases, number of opened transfer channels and excitation-energy increases. The authors proposed an expression to represent the “available energy” for a given transfer process, calculated by the change of ground-to-ground state Q-value and the change of Coulomb energy determined by identifying the outgoing two nuclides. They define that the available energy works to change the fusion reaction, due to the redistribution of the Coulomb barrier. Considering the loss of the available energy by the excitation of the system, the kinetic energy with respect to the new potential, working as the driving energy for fusion, is discussed.

The experiment is well organized, and they obtained an interesting new data with high quality and reliability. The data should be published in some journal.

However, I see a serious problem in their interpretation. They argue the impact of the MNT process on fusion. The idea that the change of the Coulomb barrier after the MNT influence the subsequent fusion is a strange argument. To realize this, fusion must happen after the MNT process finishes.

Response: This is not a **strange argument**. It has been accepted since the late 1970s that transfer reactions precede fusion, and affect fusion. The proceedings of “The Symposium on the Many Facets of Heavy Ion Fusion Reactions” published in 1986 (ANL-PHY-86-1 https://inis.iaea.org/search/search.aspx?orig_q=RN:19026117), contains 465 references to

transfer. The contribution from Dr. Rehm (p. 27 of the proceedings from 1986), starts: “Many nuclear physics textbooks treat compound nucleus formation and direct reactions [the general term for transfer-like processes] as two different entities [...]. One of the new results emerging from recent fusion reactions studies is that a close correlation exists between these two processes, in particular at energies in the vicinity of the barrier”.

Furthermore, the effect of nucleon transfer on capture/fusion has been included, in a simplified way, in models (FRESCO, CCFULL) of fusion. This is described in many reviews of fusion, such as Annu Rev Nucl Part Sci 48 (1998) 401. Transfer of nucleons is also included in models of superheavy element fusion such as the “Fusion By Diffusion” model [Swiatecki, Siwek-Wilczynska and Wilczynski Acta Physica Polonica B, 34 (2003)] and the model of Zagrebaev and Greiner [Nuclear Physics A, 944, 2015], where evolution after capture is described as a series of proton and neutron transfers. Our results are significant as they prove that this occurs with high probability outside the capture radius.

For the case of incomplete fusion, break up of weakly bound incident nucleus can happen, and simultaneously one of the clusters can fuse to target nucleus by emitting the other part of nucleus. On the other hand, in this experiment, they are looking only ejectile nuclei in the vicinity of Ca40. There might be the case that some nucleons (cluster) can go fusion after the MNT reaction, but this was not identified in the experiment by measuring evaporation residues in coincidence. Even it could happen, a nucleus slightly larger than Pb208 target is formed. I question if such a channel is important for fusion, especially for the discussion trying to understand the synthesis of SHEs.

Response: This statement highlights the key factual error in the referee’s report – following transfer of nucleons, there is no additional cluster of nucleons available to fuse with the target-like nucleus. The transfer process (as its name implies) does result in a complementary change in the target nucleus. Clusters of nucleons that may go [to] fusion (i.e. incomplete fusion) are only produced in a breakup reaction [see <https://doi.org/10.1016/j.physrep.2015.08.001>], not in multi-nucleon transfer - MNT [see <https://doi.org/10.1016/j.nimb.2013.04.093>]).

In more generally, if the MNT reactions happens before fusion, many kinds of compound nuclei are produced. If we follow the measurement and argument by the authors, the number of nuclear species as compound nuclei produced after the MNT reaction needs to be expanded significantly already at the incident energy at the Coulomb barrier.

Response: This is a factual error because if multi-nucleon transfer precedes fusion, then the same compound nucleus is formed. As an explicit example, consider a single multi-nucleon transfer channel forming ^{38}Ar and ^{210}Po (i.e. 2p stripping, $^{40}\text{Ca} + ^{208}\text{Pb} \rightarrow ^{38}\text{Ar} + ^{210}\text{Po}$). The fusion of $^{38}\text{Ar} + ^{210}\text{Po}$ that is formed in multinucleon transfer while the nuclei are approaching gives ^{248}No , exactly the same compound nucleus as that of $^{40}\text{Ca} + ^{208}\text{Pb}$ fusion. Only in a breakup process would many compound nuclei be produced.

So far, however, there is no such an experimental evidence in the measurement of evaporation residues using in-flight kinematic separator. Even at the beam energies higher than the Coulomb barrier, the produced evaporation residues can be interpreted only assuming one specific compound nucleus species, represented by the sum of protons and neutrons for projectile and target nuclei, and different evaporation channels.

Response: Measurements of binary quasifission show large yields of products heavier than the target nucleus [see, for example, Tanaka PRL 127 222501 (2021), Banerjee PRL 122

232503 (2019), Williams PRL 120 022501 (2018)]. The kinematics of MNT and quasifission makes separators extremely inefficient, but small yields are seen (A.D. Nitto et al., PLB784, 199 2018). The relevance of MNT to superheavy synthesis is explained in detail in the manuscript, which does not come from the mechanism assumed by Referee 3.

I find the authors have a serious misinterpretation in the reaction, and the argument is non-logical. Thus, I do not agree this article to be published in Nature Communications. What I can suggest is that the authors delete the discussion on the links to fusion process, and simply discuss the experimental results of MNT reaction. After the manuscript is revised, the authors would submit it in another journal devoted to nuclear physics instead of Nature Communications, as the experiment was based on the same process already established and similar experiments had been repeatedly carried out using the same devices.

Response: this conclusion seems to have arisen from the misunderstanding in the Referee report of the meaning of the term ‘transfer’ - being the movement (‘transfer’) of one or more nucleons between the two colliding nuclei - and confusing this with breakup of one nucleus.

There is extensive theoretical support for multi-nucleon transfer preceding fusion. Microscopic quantum models (TDHF and TDRPA) show that when nuclei first touch, energy dissipation and nucleon transfer between the nuclei take just a few zeptoseconds [Simenel, Godbey and Umar, PRL 212504 (2020)]. This energy dissipation and nucleon transfer is indeed what are experimentally identified as multi-nucleon transfer or deep-inelastic scattering outcomes, seen when the two nuclei re-separate. Simenel, Godbey and Umar also observe that full mass equilibration takes much longer, requiring energy and angular momentum to already be damped. This is a clear indication from microscopic models that multinucleon transfer does precede fusion.

We also note that referees 1 and 2 had no such fundamental issue with the paper. If we had made such a dire misinterpretation, the other referees (the report of referee 2 in particular seems to be that of an expert) would have noted it.

For future consideration, some of the sentence/expression is ambiguously written, so it is hard to understand even for researchers in this field. This must be improved.

Most of these comments by the referee stem from the basic misunderstandings that we have addressed above.

L36 “Superheavy elements exist at the limits of both physics and chemistry”
- *The sentence looks strange to me.*

This is difficult for us to address, as we are not sure why the referee finds this sentence strange. However, in an attempt to make it less strange, we have changed it to read “Superheavy elements exist at the extremes of physics and chemistry”

L41 “models overestimate experimental capture and fusion cross-sections.”
- *I can imagine there must be a model which underestimate experimental data. Why only models overestimating the data are discussed here.*

In fact this overestimation is a longstanding issue, with a large body of experimental and theoretical comparisons over decades. The phenomenon is universally called “suppression of

fusion” or “above barrier fusion hindrance” in the literature [See for example, the review by Dr. B.B. Back, Rev. Mod. Phys., Vol. 86 2014] because all self-consistent models of fusion overestimate the data. Only by mocking up the suppression using unphysical parameters can data be reproduced.

L42 “*explosion*”

- *What does this word means to represent collision between heavy nuclei.*

We mean a “significant and rapid increase”, in the same way someone might say “a population explosion”, which is an accepted usage of the word. The single quotes denote a figurative usage of the word, as usually done in English.

L47 “*Our interpretation overturns the current picture of fusion*”

- *The sentence is too strong. I do not agree the investigation in this article overturns the current understanding on fusion.*

Response: The basis of this statement seems to stem from the referee’s mistaken understanding outlined above. Indeed, referees 1 and 2 do not agree that this statement is too strong, with referee 2 stating “Authors find a number of very important results that challenges almost all theoretical approaches.” We believe that this sentence is appropriate.

L66 “*the identities of the nuclei are essentially unchanged prior to this point.*”

- *Concerning “the identifies of the nuclei”, what kind of properties of nucleus and the impacts on fusion reactions the authors want to argue?*

We used “identity” in the conventional sense used in physics, as in the quantities that uniquely define a nucleus – its number of proton and neutrons. To clarify this, we have changed this sentence to read “the identities of the nuclei (i.e. their proton and neutron numbers) are essentially unchanged prior to this point.”

L117 “*the smallest number of nuclide pairs making up 95% of the reflected flux rapidly increases (N95, green diamonds), reaching 31 distinct nuclide pairs.*”

- *What does the pairs mean ? One pair require two nucleons. So 31 pairs means the exchange of 62 nucleons ?*

By “nuclide pairs” we mean “pairs of nuclides”. We don’t in any way to talk about nucleons. Each nuclide measured in the reflected flux has a heavy recoiling partner – this is a pair of nuclides. For example, each ^{42}Ca we measure will be accompanied by a ^{206}Pb recoiling. To clarify this, we have changed the sentence to “the smallest number of projectile-like and target-like nuclide pairs making up 95% of the reflected flux rapidly increases (N95, green diamonds), reaching 31 distinct nuclide pairs.”

L121 “*It is not just one or two channels contributing –there is a multitude of different mass and charge transfer processes occurring. In contrast, in a typical coupled-channels calculation [1] for this system, only states in ^{40}Ca and ^{208}Pb and a few simple transfer reactions would be included.*”

- *I guess to explain the current experimental data of $\text{Ca}40+\text{Pb}208$ MNT reaction is already outside the scope in the model of [1]. So, this sentence can be deleted.*

Response: We disagree. The contrast between what is measured in the reflected flux occurring outside the barrier compared to what is modelled in coupled channels is highly relevant. Since at deep-sub barrier energies there is essentially negligible absorbed flux, the

reflected flux represents all outcomes. The outcomes we measure are not those included in coupled channels calculations of capture.

Furthermore, there have been theoretical attempts to describe MNT processes using extensions to the coupled channels formalism [Scamps & Hagino, PRC 92 054614 (2015)], demonstrating that these experimental data are not “outside the scope” of coupled channels models.

L134 *“The excitation energy is largest when multiple nucleons are transferred, with $\langle Ex \rangle = 29.5$ MeV, though even the inelastic and one & two nucleon transfer component shows a mean excitation energy of $\langle Ex \rangle = 10.0$ MeV”*

- *In Fig.1(c), there is no data point showing average excitation energy $\langle Ex \rangle = 29.5$ MeV. What does it mean ?*

Thank you for the opportunity to clarify this. Fig 1(c) shows the mean excitation energy for the whole of the reflected flux, which is discussed in the previous sentences to the one the referee quotes. The quoted sentence refers to the mean excitation energies for subsets of these data, when multiple nucleons are transferred ($\langle Ex \rangle = 29.5$ MeV) and when zero to two nucleons are transferred ($\langle Ex \rangle = 10.0$ MeV). To make this point clearer, we have modified the text to include a specific reference to Fig 5 (b) and (c) in the extended data, which shows these excitation energies.

L177 *“Significant energy loss (up to hundreds of MeV), associated with complex multinucleon transfers in both directions, is known as deep-inelastic scattering [14, 24, 25].”*

- *What does “in both directions” mean ?*

We mean “transfer of nucleons towards the 208Pb target and away from the 208Pb target”. This is accepted nomenclature in the field [see as an example, Sekizawa and Yabana, PRC 88 014614 (2013), pg 9, first paragraph]. However, we agree that this might not be clear for the non-expert reader. We have hence changed this sentence to read “Significant energy loss (up to hundreds of MeV), associated with complex multinucleon transfers both towards and away from the target, is known as deep-inelastic scattering [14, 24, 25].”

L194 *“How high does the excitation energy need to be?”*

- *This sentence is strange.*

Again, it is difficult for us to address this point as the referee has not told us what they find strange about this sentence. We have changed it to read “How high does the excitation energy need to be to lead to (effective) irreversibility?”

L270 *“one would expect enhanced fusion and at least a 10 MeV wide fusion barrier distribution.”*

- *I do not agree this argument. The discussion is only for the MNT process. How can we argue if the remaining nucleons after ejecting Ca40-like nuclei can go fusion ?*

Response; This is evidence that the referee report has confused breakup with multinucleon transfer. This statement: “How can we argue if the remaining nucleons after ejecting Ca40-like nuclei can go fusion?” requires that there are some “remaining nucleons”, which would only occur if the 40Ca had broken up, rather than transferred nucleons. Multi-nucleon transfer (moving nucleons from one nucleus to another) does not leave remaining nucleons.

L279 *“Significantly, there is also an exponentially falling tail extending at least as far as $\Delta E_{fi} = -40$ MeV, that will reduce fusion.”*

- *Again from this experimental data, one cannot say anything on fusion process.*

Response: Our measurements were made for collisions with distances of closest approach outside the capture barrier so that we can observe all the products of MNT. Clearly if 40 MeV of kinetic energy is dissipated, to fuse, an additional 40 MeV of beam energy will be required. Our argument of how this must affect fusion comes from basic energy conservation. Only by asserting breakdown of energy conservation could we be prevented from drawing conclusions about fusion.

L330 *“Before capture, multinucleon transfer results in a distribution of Z1Z2.”*

- *Again there is no experimental evidence that capture process happens after multinucleon transfer (MNT) process finishes. The authors measured only MNT ejectiles and nothing is measured to argue the evidence of subsequent fusion.*

Response: our measurements are made for collisions with distances of closest approach outside the capture barrier so that we can observe all the products of MNT. At above-barrier energies, reaching inside the capture barrier radius, fusion/quasifission does occur. Since we find almost all flux outside the barrier has undergone MNT, if capture/fusion could not follow MNT, there would be essentially no fusion observed, in disagreement with experimental cross sections.

L327 *“Our observations of the multitude of identities mean that each of these variables may be changed enroute to capture.”*

- *Again it is hard to believe fusion and quasifission processes are influenced by the MNT reaction. They are competing and independent process. Two-step process involving the MNT and fusion/quasifission has not been identified experimentally.*

Response: the referee should not assert that MNT and fusion/quasifission are both “competing and independent process”. In fact, multinucleon transfer and quasi-fission have recently been shown to evolve smoothly from one another [see, for example, Tanaka PRL 127 222501 (2021), Banerjee PRL 122 232503 (2019), Williams PRL 120 022501 (2018)]. These experiments indicate clearly that multinucleon transfer and quasifission make up a continuum of outcomes resulting from increasing energy dissipation and mass evolution with increasing time, mediated by the same mechanism of nucleon transfers.

What can be changed is only their cross sections due to the loss of incident flux by the MNT process.

Response: MNT does not result in reduction of incident flux, unlike breakup, which the referee is evidently confusing it with.

Fig.3 The authors show the distribution of the yield of ejectile nuclei. This is the results of MNT reaction, and this does not mean fusion/quasifission can happen from these different Z1Z2 combinations.

Response: there is a large body of evidence for competition between fusion and faster processes such as quasifission [see recent review in Progress in Particle and Nuclear Physics 118, 103856 (2021)]. Also TDHF model calculations indicate that in the first \sim zs of collision, multi-nucleon transfer takes place, followed by quasifission and fusion. Since only a few percent of the flux at the barrier is not in different Z1Z2 combinations,

fusion/quasifission must occur from these different combinations, to match experimental cross sections.

L582 “The overall shape of the distributions will not be substantially moderated by the acceptance of PRISMA, except at the edges of the acceptance, reached only at the higher beam energies, thus allowing qualitative comparisons of their evolution with energy.”

- The meaning is not clear to me.

In order to clarify this sentence, we have changed it to read: “The efficiency of PRISMA is quite uniform, except for at the edges of its acceptance [54]. Therefore, the overall shape of the measured distributions are not substantially moderated by acceptance effects, thus allowing qualitative comparisons of their evolution with energy.”

Fig.4 (caption) “Diagonal lines indicate the isospin asymmetry equal to that of the compound nucleus ^{248}Fm ($N/Z = 1.45$)”

- N/Z for ^{248}Fm is 1.48 (not 1.45).

Thank you for bringing this typo to our attention! The compound nucleus is ^{248}No , not ^{248}Fm , and the correct N/Z is 1.43. This has been corrected. The lines in the figure were correctly drawn for $N/Z=1.43$, and hence they have not been changed.

REVIEWERS' COMMENTS

Reviewer #1 (Remarks to the Author):

Dear authors;

According to my suggestions, the author has made great improvements to the manuscript of the article and agrees to publish the current manuscript in Nature communications.

Sincerely yours;

Reviewer #2 (Remarks to the Author):

I have read the response to the questions/comments I have raised and the authors answered them and made the necessary changes to my satisfaction.

I have also read the comments and the author's responses to the comments made by referee's #1 and #3. I would like to add some comments regarding some of those comments.

Referee #1, questions the influence of angular momentum on the Z, N and E_x . I agree with the author's response to this question.

Microscopic dynamical calculations show that l -dependence of the fusion barrier (by directly calculating fusion potentials for non-central collisions) is minimal and the cross-sections primarily depend on the addition of centrifugal potential to the main barrier.

Referee #2, I agree with author's responses to the referee's comments. Unfortunately, as our fields become more and more specialized certain terminologies carry different meanings in different subfields. The questions seem to come from the reaction theory perspective of ion-ion collisions. In the heavy-ion fusion and related reactions it is well known that MNT reactions precede fusion and actually they open and facilitate the pathway to fusion. If we think of two nuclei approaching each other in a time-dependent fashion, first a neck is formed when the nuclear wavefunctions start to overlap. This doorway leads to MNT and MNT leads to eventual capture. For energies well below the barrier this may happen more slowly but something has to happen to bring these nuclei to the point of capture (enough nuclear attraction). Naturally, this is a low probability event for deep sub-barrier collisions indicated by low cross-sections but trying to understand or experimentally determine the details of this process is fundamentally important to our understanding of many-body tunneling and fusion.

November 8 2023.

Response to referees

We thank the referees for their efforts in reviewing this manuscript. Since referee 1 has raised no issues to address, we do not include a point-by-point response.

We thank Referee 2 for their comments. It is absolutely important to describe heavy ion collisions as a time dependent process, as they well describe. We ultimately agree that nomenclature may be to blame for a miscommunication with Referee 3. This is another lesson that as a wider field, it is important that we try to maintain some common language – clearly the field of heavy ion fusion has benefited from ideas from ion-ion collisions over decades, and vice-versa. This is something to keep in mind, and we appreciate Referee 2 for bringing it up!

Best,

Kaitlin Cook
(on behalf of the Authors).

Reviewer #1 (Remarks to the Author):

Dear authors;

According to my suggestions, the author has made great improvements to the manuscript of the article and agrees to publish the current manuscript in Nature communications.

Sincerely yours;

Reviewer #2 (Remarks to the Author):

I have read the response to the questions/comments I have raised and the authors answered them and made the necessary changes to my satisfaction.

I have also read the comments and the author's responses to the comments made by referee's #1 and #3. I would like to add some comments regarding some of those comments.

Referee #1, questions the influence of angular momentum on the Z,N and E_x . I agree with the author's response to this question. Microscopic dynamical calculations show that l -dependence of the fusion barrier (by directly calculating fusion potentials for non-central collisions) is minimal and the cross-sections primarily depend on the addition of centrifugal potential to the main barrier.

Referee #2, I agree with author's responses to the referee's comments. Unfortunately, as our fields become more and more specialized certain terminologies carry different meanings in different subfields. The questions seem to come from the reaction theory perspective of ion-ion collisions. In the heavy-ion fusion and related reactions it is well known that MNT reactions precede fusion and actually they open and facilitate the pathway to fusion. If we think of two nuclei approaching each other in a time-dependent fashion, first a neck is formed when the nuclear wavefunctions start to overlap. This doorway leads to MNT and MNT leads to eventual capture. For energies well below the barrier this may happen more slowly but something has to happen to bring these nuclei to the point of capture (enough nuclear attraction). Naturally, this is a low probability event for deep sub-barrier collisions indicated by low cross-sections but trying to understand or experimentally determine the details of this process is fundamentally important to our understanding of many-body tunneling and fusion.